A taxonomic reassessment of Piramys auffenbergi, a neglected turtle from the late Miocene of Piram Island, Gujarat, India

http://orcid.org/0000-0003-1554-8346 Ferreira Gabriel S. 1 2 gsferreira@usp.br
Bandyopadhyay Saswati 3
Joyce Walter G. 4 walter.joyce@unifr.ch
1 Faculdade de Filosofia, Ciências e Letras de Ribeirão Preto, Universidade de São Paulo , Ribeirão Preto , Brazil
2 Fachbereich Geowissenschaften, Eberhard-Karls-Universität Tübingen , Tübingen , Germany
3 Geological Studies Unit, Indian Statistical Institute , Kolkata , India
4 Departement für Geowissenschaften, University of Fribourg , Freiburg , Switzerland
Abdala Virginia
Electronic publication date: 2018 Nov 15
Publication date: 2018
Volume: 6
Electronic Location ID: e5938
Received 2018 Aug 17; Accepted 2018 Oct 15
Copyright: © 2018 Ferreira et al.
Copyright year: 2018
Copyright holder: Ferreira et al.
License: This is an open access article distributed under the terms of the Creative Commons Attribution License, which permits unrestricted use, distribution, reproduction and adaptation in any medium and for any purpose provided that it is properly attributed. For attribution, the original author(s), title, publication source (PeerJ) and either DOI or URL of the article must be cited.
License URL: https://creativecommons.org/licenses/by/4.0/

Keywords: Taxonomy, Late Miocene, Stereogenyini, Pleurodira, Piram Island, India

Funding: The Swiss National Science Foundation SNF 20021_153502/2 The Fundação de Amparo à Pesquisa do Estado de São Paulo (FAPESP) 2014/25379-5 and 2016/03934-2 This contribution was funded by grants from the Swiss National Science Foundation to Walter G. Joyce (SNF 20021_153502/2) and the Fundação de Amparo à Pesquisa do Estado de São Paulo (FAPESP) to Gabriel S. Ferreira (2014/25379-5 and 2016/03934-2). The funders had no role in study design, data collection and analysis, decision to publish, or preparation of the manuscript.

==============================
Background

Piramys auffenbergi was described as an emydine turtle based on a well-preserved skull retrieved from late Miocene deposits exposed on Piram Island, India. The description and figures provided in the original publication are vague and do not support assignment to Emydinae. This taxon has mostly been ignored by subsequent authors.

Material and Methods

We reexamine the holotype specimen, provide an extensive description and diagnosis for Piramys auffenbergi, and include this taxon in a global character-taxon matrix for Pleurodira.

Results

The presence of a processus trochlearis pterygoidei conclusively shows pleurodiran affinities for Piramys auffenbergi. Inclusion of this taxon in a phylogenetic analysis retrieves it within Stereogenyini closely related to the Asian taxa Shweboemys pilgrimi and Brontochelys gaffneyi.

Discussion

Our reexamination of the holotype of Piramys auffenbergi confidently rejects the original assessment of this taxon as an emydine testudinoid and conclusively shows affinities with the pleurodiran clade Stereogenyini instead. Even though most taxa from this lineage are thought to be coastal turtles, all Asian stereogenyines were collected from continental deposits, suggesting a more diverse paleoecology for the group.

Introduction

Piramys auffenbergi was described by Prasad (1974) based on a single skull (GSI 18133) retrieved from Neogene sediments exposed on the small island of Piram, Gujarat State, India (Fig. 1). Prasad (1974) identified the taxon as an emydine, but the diagnosis and the description are general and do not reveal any emydine affinities. Today, only cryptodiran cheloniids, geoemydids (formerly part of Emydidae), testudinids, and trionychids occur in India (Turtle Taxonomy Working Group (TTWG), 2017), but stereogenyine pleurodires inhabited the area as recently as the Plio/Pleistocene (Gaffney et al., 2011).

Figure 1 The type locality of Piramys auffenbergi.

(A) A simplified map of India. Gujarat State is highlighted in purple, all other states in various shades of gray. (B) Detailed view of the Gulf of Khambhat. Piram Island is highlighted in purple.

Here, we provide an extensive description and diagnosis of Piramys auffenbergi based on a reexamination of the holotype specimen. Phylogenetic analysis concludes that this taxon is a representative of Stereogenyini, a clade that includes other South Asian forms, namely the Plio/Pleistocene Shweboemys pilgrimi Swinton, 1939 from Myanmar, and the late Oligocene Brontochelys gaffneyi (Wood, 1970) from Pakistan. We also provide an updated account of the paleoecology and biogeography of the group, concluding that, although they were most likely adapted (or at least highly tolerant) to salty waters, stereogenyines were still restricted geographically and not as widespread as modern sea turtles.

Systematic Paleontology

PLEURODIRA Cope, 1865

PELOMEDUSOIDES Broin, 1988

PODOCNEMIDIDAE Cope, 1868

STEREOGENYINI Gaffney et al., 2011

STEREOGENYITA Gaffney et al., 2011

Piramys auffenbergi Prasad, 1974

Holotype. GSI 18133, a partial skull (Prasad, 1974, figs. 2–4, pl. 3.1; Fig. 2).

Figure 2 GSI 18133, Piramys auffenbergi, holotype, late Miocene of Piram Island, Gujarat, India.

Photographs and illustrations of specimen in (A) dorsal, (B) right lateral, (C) left lateral, (D) anterior, and (E) posterior views. Abbreviations: cm, condylus mandibularis; et, Eustachian tube; fnt, foramen nervi trigemini; fr, frontal; fst, foramen stapedio-temporale; ju, jugal; mx, maxilla; oc, occipital condyle; op, opisthotic; pa, parietal; pf, prefrontal; po, postorbital; pr, prootic; ptp, processus trochlearis pterygoidei; qj, quadratojugal; qu, quadrate; so, supraoccipital; sq, squamosal.

Type locality and horizon. Piram Island (Fig. 1), Gulf of Khambhat (formerly Gulf of Cambay), Gujarat, India; conglomerate beds, middle Siwaliks (Prasad, 1974), Dhok Pathan age, late Miocene (Nanda, Sehgal & Chauhan, 2017).

Referred material. No specimens have been referred to date.

Diagnosis. Piramys auffenbergi can be diagnosed as a representative of Pleurodira by the presence of the processus trochlearis pterygoidei and of Pelomedusoides by the absence of nasal bones. Piramys auffenbergi can be further diagnosed as a representative of Stereogenyini by the small entrance to the antrum postoticum, the condylus mandibularis projecting ventrally from the cavum tympani, and the posteriorly broad skull, and as a member of Stereogenyita closer to Shweboemys pilgrimi and Stereogenys cromeri by a median notch in the upper jaw. Piramys auffenbergi is distinguished from the other members of Stereogenyini by the following combination of characters: a pinched snout (not present in Latentemys plowdeni Gaffney et al., 2011, Bairdemys venezuelensis Wood & Díaz de Gamero, 1971, and Bairdemys sanchezi Gaffney et al., 2008), a deep interorbital groove on the prefrontal and frontal (otherwise present in Cordichelys antiqua Andrews, 1903), foramen nervi trigemini located above the level of the sulcus palatinopterygoideus (in contrast to the more ventrally displaced foramen nervi trigemini in Shweboemys pilgrimi and Brontochelys gaffneyi), the lower temporal emargination rising above the ventral level of the orbit (shallower in C. antiqua and unknown for other Stereogenyita), and the foramen stapedio-temporale placed anteriorly (autapomorphic character).

Description

Preservation. The skull was likely buried intact but suffered much weathering damage more recently. The bone is nevertheless of good quality and sutures are well preserved. The palate and ventral aspects of the basicranium are still obscured by matrix, but the dermatocranium and the dorsal parts of the basicranium are exposed (Fig. 2). The margins of the external nares and the median portions of the labial margins are damaged, but the remaining margins likely approximate their original outlines. The temporal roofing, by contrast, is massively eroded and the extent of the upper temporal emarginations can therefore not be assessed anymore. Much of the postorbitals, parietals, quadratojugals, squamosals, and the supraoccipital are missing. Faint scute sulci are apparent on the postorbitals and parietals (Fig. 2A).

Prefrontal. The prefrontals are relatively small, paired elements that form the anterodorsal aspects of the orbit and the anterior third of the narrow interorbital space. The anterior margins are damaged, but the prefrontals likely formed the dorsal roof of the external nares, as nasals were absent. The prefrontals contact the frontals posteriorly along a straight, transverse suture, the maxillae ventrally along a straight, horizontal suture, and one another along a straight, median suture. The descending branch of the prefrontal contacts the frontal posteriorly and the maxilla ventrally, but potential, more-distal contacts with the vomer and palatine are obscured by matrix (Fig. 2C). The prefrontals jointly form a deep, median grove on the dorsal skull surface (Fig. 2A).

Frontal. The frontals are the only bones that are undamaged. The dorsal plate of the frontals forms the posterior two thirds of the interorbital space and posterodorsal margins of the anterolaterally facing orbits. The frontals contact the prefrontals anteriorly along a straight, transverse suture, the postorbitals laterally along parasagittal sutures of uneven length, the parietals posteriorly along straight, transverse sutures, and one another along a straight, median suture. Within the orbits, the frontals contact the prefrontals anteriorly and the postorbitals laterally. The median groove formed by the prefrontals continues posteriorly onto the frontals (Fig. 2A), but fades to oblivion towards the posterior margins of these elements.

Parietal. Symmetric damage to the dorsal plate of the parietal creates the illusion that the parietals are relatively small elements that define the margins of well-developed upper temporal emarginations, but all margins show signs of damage and the extent of this emargination cannot be assessed with any confidence. It is therefore only clear that the dorsal plates of the parietals contact the frontals anteriorly along straight, transverse sutures and the postorbitals along oblique, anterolateral sutures. Together with the postorbitals, the parietals form a pair of sulci that converge towards the posterior and that likely trace the outline of a large internarial scute (scute ii of Ferreira et al., 2015) that covered the prefrontals, frontals, and the medial portions of the postorbitals and parietals. The descending process of the parietals contacts the pterygoid anterior to the foramen nervi trigemini, forms the dorsal margin of this foramen, and contacts the prootic and supraoccipital posterior to it (Fig. 2A). The foramen nervi trigemini is located above the level of the sulcus palatinopterygoideus and is well separated from the foramen stapedio-temporale by the prootic (Fig. 2C). More anterior contacts, if present, are obscured by matrix.

Postorbital. The posterior margins of the postorbitals show signs of damage and it is therefore not possible to assess if they originally contributed to the upper temporal emarginations, although the thickness of their damaged margins makes this implausible. However, as the intact posterior margins converge, it is unlikely that they were significantly larger than preserved. The dorsal plates of the postorbitals at least contact the frontals medially, the parietals posteromedially, the jugals ventrally, and the quadratojugals posterolaterally. The descending process of the postorbital forms much of the posterior wall of the orbit, but its distal contacts are obscured by matrix. We are not able to discern a dorsal pocket within the upper limits of the posterior wall of the orbit.

Jugal. The jugals are relatively large elements that form the posteroventral aspects of the orbit. Although the posteroventral aspects are damaged on both sides, enough is preserved to indicate that a well-developed lower temporal emargination is developed that rises to the level of the lower third of the orbit and that is anterodorsally framed by the jugals (Fig. 2B). The jugals contact the postorbitals dorsally along horizontal sutures, the quadratojugal posteriorly along a transverse suture, and the maxillae anteroventrally along a slightly oblique suture. The jugals contact the maxilla anteriorly and the postorbitals laterally on the surface of the skull, but matrix obscures possible contacts within the orbit.

Quadratojugal. Only a fragment of the quadratojugal remains on the right side of the skull (Fig. 2B). It is therefore unclear if the quadratojugal contributes to the upper or lower temporal emarginations, although a contribution to the lower temporal emargination appears highly likely. Only a short, anteromedial contact is preserved with the postorbital, an elongate anterior contact with the jugal, and remnants of the posterior contact of the quadrate anterior to the cavum tympani.

Squamosal. A fragment of the right squamosal is available at the back of the skull (Fig. 2A). This fragment contacts the quadrate anterolaterally and the opisthotic anteromedially within the upper temporal fossa. A likely contribution of the squamosal to the region posterior to the cavum tympani cannot be discerned from that of the quadrate. However, the massive nature of the damaged quadrates suggests that the squamosal did not contribute to the antrum postoticum.

Premaxilla. The anterior tip of the skull is damaged, but the median, cleft like notch appears to be genuine, as much of the left labial margin is intact (Fig. 2D). We are unable to discern the contacts, or even the presence, of premaxillae.

Maxilla. The maxillae can only be observed in anterior and lateral views, as the palate is hidden by matrix. On the skull surface, the high maxillae contact the jugals below the orbit along transverse sutures and the prefrontals dorsally along horizontal suture. These contacts can partially be traced within the orbits, but matrix obscures their full extent. It is unclear if an anterior contact is developed with the premaxillae. Likely contacts with the palatines are obscured by matrix. The maxillae otherwise form the labial margins of the jaw, the anteroventral margins of the orbit, and the lateral margin of the broad external nares.

Pterygoid. Only the most lateral and medial tips of the left pterygoid are visible (Fig. 2A). Medially, the pterygoid forms the anterior margin of the foramen nervi trigemini and contacts the parietal dorsally. Laterally, the pterygoid forms the processus trochlearis pterygoidei, of which only the damaged lateral tip is free from matrix (Fig. 2C). The process is perpendicular to the midline and almost vertical in lateral view.

Prootic. Only the left prootic can be observed in dorsal view and its sutures are obscured by localized damage to the surface of the skull. The left prootic clearly contacts the descending process of the parietal anteromedially and the quadrate laterally. Contacts are certainly present with the supraoccipital posteromedially and the opisthotic posteriorly, but their orientation and size cannot be estimated with confidence. The left prootic partially roofs the foramen nervi trigemini and together with the left quadrate forms the foramen stapedio-temporale, which is situated at the anterior margin of the otic capsule and oriented anteriorly (Fig. 2A).

Opisthotic. Only the external aspects of the opisthotics can be observed. Within the upper temporal fossa, the opisthotics contact the supraoccipital anteromedially, the quadrate anterolaterally, the squamosal posterolaterally, and the exoccipital posteromedially.

Quadrate. Much of the quadrates remain on both sides of the skull, but most surfaces are damaged, making it difficult to discern its original shapes and contacts. At the very least, the quadrate contacts the quadratojugal anterodorsally and the squamosal posteriorly along the skull surface. Within the upper temporal fossa, the quadrate contacts the squamosal posteriorly, the opisthotic posteromedially, and the prootic anteromedially. Although damage to the middle ear is significant, it is apparent that the quadrates fully enclosed the incisura columella auris and Eustachian tube and that the quadrates fully formed the highly reduced antrum postoticum. However, matrix obscures the presence of a possible precolumellar fossa. In lateral view (Fig. 2C), the mandibular condyle of the quadrate projects ventrally, beyond the ventral outline of the cavum tympani.

Supraoccipital. The supraoccipital is heavily eroded and some of its contacts are obscured by damage. It is therefore not possible to assess its possible contribution to the dorsal skull roof or the length of the supraoccipital crest. Within the upper temporal fossa, the supraoccipital certainly contacts the parietal anteromedially, the prootic anterolaterally, the opisthotic posterolaterally, and the exoccipital posteriorly. In posterior view, the supraoccipital roofs the foramen magnum and contacts the exoccipitals ventrally.

Exoccipital. The exoccipitals are both present, but their ventral portions are obscured by matrix. These bones form the lateral margins of the foramen magnum, contact the supraoccipital dorsally, the opisthotic anterolaterally, but their likely ventral contacts with the basioccipital are obscured. Although the occipital condyle is heavily damaged, the sulcus left by the contact between the exoccipitals laterally and the basioccipital ventrally are still visible (Fig. 2E).

Material and Methods

We integrated the holotype of Piramys auffenbergi into the phylogenetic matrix of Ferreira et al. (2018) to assess its relationships to other pleurodires. Obtaining CT scans of this specimen would have been desirable, as this would help clarify the morphology of the palate, but was not feasible within the context of this study due to logistic hurdles. The matrix was modified through the addition of a new character (ch. 245: PM, median notch on upper jaw; see Supplementary Files) and by updating the scorings of some taxa based on new observations by one the authors (GSF) (see Supplementary Files for full list of changes). As in the original analysis (Ferreira et al., 2018), twelve characters (14, 18, 19, 71, 95, 96, 99, 101, 119, 129, 174, 175) were interpreted as forming morphoclines and were ordered in the analysis. The resulting matrix was analyzed in TNT v. 1.5 (Goloboff & Catalano, 2016) using a traditional search with 2,000 replicates of Wagner trees, random seed set to 0, branch-swapping algorithm tree bisection reconnection (TBR), hold = 0, and collapsing zero-length branches according to rule “1”. The most parsimonious trees (MPTs) were subject to a second round of TBR and a strict consensus was obtained from the resulting MPTs. Consistency (CI) and Retention (RI) indexes, Bremer support, and resampling values (bootstrap and jackknife, using 1,000 resamples for calculating absolute and difference of frequencies; Goloboff et al., 2003) were retrieved using implemented functions in TNT.

Results

The search yielded 270 MPTs with 1,134 steps (CI = 0.288; RI = 0.748). The strict consensus (see Supplementary Files) differs slightly from the results of Ferreira et al. (2018) by yielding a polytomy in the clade that includes the extant Erymnochelys madagascariensis (Grandidier, 1867) and Peltocephalus dumerilianus (Schweigger, 1812). Piramys auffenbergi is retrieved with relatively high support (Bremer support = 2; see Supplementary Files) inside Stereogenyini, in a polytomy with Stereogenys cromeri Andrews, 1901 and Shweboemys pilgrimi (Fig. 3).

Figure 3 Phylogenetic hypothesis of Stereogenyina.

A time-calibrated cladogram depicting a portion of the strict consensus topology of 270 MPTs with 1,134 steps retrieved from the phylogenetic analysis using ordered characters. Dark lines highlight the known temporal distribution of a taxon and the colored continent symbols highlight their known spatial distribution. The full tree is provided in the File S2.

Discussion

Alpha taxonomy and phylogenetic relationships

Prasad (1974) documented in the original description and figures of the holotype of Piramys auffenbergi the presence of large nasals, elongate prefrontals, small frontals that do not contribute to the orbits, and deep upper temporal emarginations. Prasad (1974) unfortunately did not explain why he felt Piramys auffenbergi to be an emydine, but we note that the presence of nasals is neither consistent with an assignment to Emydinae in particular, nor Testudinoidea more generally, as testudinoids universally lack nasal bones (Gaffney, 1979). Our re-examination of the type specimen allows us to better evaluate the cranial morphology and taxonomic affinities of this taxon and to present a more detailed list of diagnostic characters.

The most apparent differences in our interpretation of the skull is that nasals are absent, that the prefrontals are situated at the front of the orbits, that the frontals broadly contribute to the orbits, that the margin of the upper temporal emargination is not preserved, and that a trochlear process is present on the pterygoid. The presence of a trochlear process of the pterygoid combined with the absence of nasals clearly hint at pelomedusoid affinities for Piramys auffenbergi (Gaffney, 1979; Gaffney, Tong & Meylan, 2006; Joyce, 2007). The trochlear process is almost completely embedded in matrix, but its lateral tip can be seen anterior to the otic capsule (Fig. 2C).

The posteriorly broadened skull, small anterior opening of the antrum postoticum, and a mandibular condyle projecting ventrally from the cavum tympani in lateral view (Fig. 2C) suggest stereogenyine affinities (Gaffney et al., 2011; Ferreira et al., 2015). The results of our phylogenetic analysis (Fig. 3) support this initial hypothesis, even though Piramys auffenbergi is scored as “unknown” for several characters that support the monophyly of Stereogenyini. The majority of characters that resolve relationships within Stereogenyini, for example, presence of a secondary palate with a midline cleft (Gaffney et al., 2011; Ferreira et al., 2015), refer to features in the ventral region of the skull. Unfortunately, this region is covered by matrix in GSI 18133, hampering a more detailed account of its taxonomic affinities. Nevertheless, the pinched snout (ch. 41, state 1), premaxillae protruding anteriorly beyond the dorsal edge of the apertura narium externa (ch. 43, state 1), the small opening of the antrum postoticum (ch. 86, state 2), the ventral projection of the mandibular condyle (ch. 90, state 1), and the median notch in the upper jaw (ch. 246, state 2) supports Piramys auffenbergi as a Stereogenyini with close affinities to Stereogenys cromeri and Shweboemys pilgrimi (Fig. 3).

Comparisons of Piramys auffenbergi with other Asian stereogenyines reveal osteological and chronological distinctness. The Bugti Hills in Pakistan, the type locality of Brontochelys gaffneyi, are now considered to be late Oligocene in age (Welcome et al., 2001; see Discussion in the next section) whereas the Irrawaddy Beds in Myanmar, the type locality of Shweboemys pilgrimi, are thought to be Pliocene/Pleistocene age (Thein, Nu & Pyone, 2012). The sediments exposed on Piram Island are dated as late Miocene (Nanda, Sehgal & Chauhan, 2017) and Piramys auffenbergi therefore is temporally distinct. Piramys auffenbergi differs from Brontochelys gaffneyi by the presence of a pinched snout, anterodorsally facing orbits, and a median notch in the upper jaw. On the other hand, Piramys auffenbergi differs from Shweboemys pilgrimi by the presence of an interorbital space that is narrower than the diameter of the orbits and the straight prefrontal-frontal suture. Furthermore, Piramys auffenbergi can be distinguished from both other Asian stereogenyines by the presence of a narrow interorbital groove, the more posterodorsally located foramen nervi trigemini, and the anterior, instead of dorsal, opening of the foramen stapedio-temporale. We are therefore able to confirm the validity of Piramys auffenbergi, though not as a testudinoid cryptodire, but rather as a podocnemidid pleurodire.

Paleoecology and biogeography

Members of Stereogenyini are currently thought to be durophagous, because they have broad and flat triturating surfaces combined with a secondary palate (Ferreira et al., 2015), features that are prevalent among extant turtles with durophagous dietary preferences (Foth, Rabi & Joyce, 2017). The palate is not exposed in the holotype of Piramys auffenbergi and we are therefore not able to assess the diet of this taxon with confidence. However, the high maxillae, anterodorsally oriented orbits, and posteriorly broadened skull with space for well-developed adductor muscles (Fig. 3) are features consistent with this life style. Considering the placement of Piramys auffenbergi within Stereogenyini we therefore predict the presence of durophagous features in this taxon as well.

Members of Stereogenyini are furthermore thought to have had a marine, or at least coastal, lifestyle based on limb morphology (Weems & Knight, 2013), shell morphology (Pérez-García, 2017), and depositional environments (Ferreira et al., 2015). Although most stereogenyine taxa have indeed been recovered from sediments that represent coastal or open sea depositional environments (Ferreira et al., 2015), we here note that all known Asian stereogenyines were collected from continental deposits, at least as indicated by the prevalence of terrestrial mammals in combination with fresh water aquatic fish and reptiles. In particular, the type of Brontochelys gaffneyi was collected from late Oligocene (not Miocene as reported Wood, 1970; Gaffney et al., 2011) continental sediments exposed in the region of Dera Bugti, Pakistan (Marivaux, Vianey-Liaud & Welcomme, 1999; Welcome et al., 2001), Shweboemys pilgrimi from the continental Plio/Pleistocene Irrawaddy Beds of Myanmar (Thein, Nu & Pyone, 2012), and, as demonstrated herein, Piramys auffenbergi from the continental late Miocene middle Siwaliks of India (Nanda, Sehgal & Chauhan, 2017). However, the more marine adapted African representatives of the Tethyan clade Stereogenyita, in particular Stereogenys cromeri and Lemurchelys diasphax, indicate a certain amount of ecological fluidity within the group. Bothremydids, another clade of pelomedusoid pleurodires, also present a combination of marine and continental forms (Rabi, Tong & Botfalvai, 2012), suggesting that a high tolerance to saline waters may be a more widespread feature among side-necked turtles. This is also hinted at by experimental analyses (Bower et al., 2016). However, as deltaic and marine sediments often interfinger finely and as estuarine and coastal turtles are expected to be found in numerous coastal facies, we suggest that the available data is not sufficient to establish a rigorous pattern for the moment. Nevertheless, the fact that our phylogenetic analysis retrieves two geographically separated clades, the Tethyan Stereogenyita and the South American Bairdemydita, suggests that the even more salt tolerant forms were not highly marine, as greater amounts of dispersal and less endemism were otherwise to be expected.

Despite the broad geographic distribution of stereogenyines across North and South America, Africa, and India, they did not achieve the degree of cosmopolitanism seen in modern sea turtles (Chelonioidea; Turtle Taxonomy Working Group (TTWG), 2017). Instead, it appears that they were restricted to equatorial and subtropical regions of the northern hemisphere (Fig. 4). This hints at the possibility that their distribution was limited by cooler temperatures outside of the tropics.

Figure 4 Geographic distribution of stereogenyines.

Section of world map indicating localities of extinct taxa assigned to Stereogenyini. Red circles indicate Stereogenyita, green circles Bairdemydita, and blue circles non-Stereogenyita and non-Bairdemydita steregeonyines. 1, Bairdemys venezuelensis; 2, B. thalassica Ferreira et al., 2015; 3, B. sanchezi; 4, B. winklerae Gaffney et al., 2008; 5, B. hartsteini Gaffney & Wood, 2002; 6, B. healeyorum Weems & Knight, 2013; 7, Mogharemys blanckenhorni (Dacqué, 1912); 8, Cordichelys antiqua; 9, Latentemys plowdeni; 10, Stereogenys cromeri; 11, Lemurchelys diasphax Gaffney et al., 2011; 12, Brontochelys gaffneyi; 13, Shweboemys pilgrimi; 14, Piramys auffenbergi. Abbreviations: EG, Egypt; MM, Myanmar; PK, Pakistan; PR, Puerto Rico; US, United States of America; VE, Venezuela.

Conclusions

Our re-examination of the type specimen revealed the presence of pelomedusoid features in Piramys auffenbergi, for example, presence of a pterygoid trochlear process in combination with the absence of nasals. Another set of characters suggested a Stereogenyini affinity which was confirmed by the results of our phylogenetic analysis of a global matrix for Pleurodira. Piramys auffenbergi is nested closer to the other Asian stereogenyines Brontochelys gaffneyi and Shweboemys pilgrimi. The phylogenetic position and provenance of Piramys auffenbergi is in accordance with the Stereogenyini broad geographic distribution and proposed high dispersal capability, but the relations within this lineage, with a South American and a Tethyan clade, suggest they were not as cosmopolitan as modern sea turtles.

Supplemental Information

Supplemental Information 1 Character taxon matrix used in phylogenetic analysis, including full character list and character state definitions.

Click here for additional data file.

Supplemental Information 2 List of changes to Ferreira et al. (2018)’s matrix, strict consensus tree retrieved from the phylogenetic analysis, bootstrap, jackknife and Bremer support values, and list of common synapomorphies for Stereogenyina.

Click here for additional data file.

We would like to thank all officials and supporting staff at the Curatorial Division of the Geological Survey of India and of the Indian Museum for generously providing access to collections under their care, in particular Senior Geologist A. Bhattacharya, Senior Geologist M. R. Moulick, and Senior Geologist A. K. Mondal. The editor, Virginia Abdala, and the reviewers, Juliana Sterli, Márton Rabi and Torsten Scheyer, are thanked for insightful comments that helped improve the quality of the manuscript.

Institutional Abbreviation

GSI Geological Survey of India, Kolkata, India.

Additional Information and Declarations

Competing Interests

Author Contributions

Data Availability

The authors declare that they have no competing interests.

Gabriel S. Ferreira analyzed the data, prepared figures and/or tables, authored or reviewed drafts of the paper, approved the final draft.

Saswati Bandyopadhyay authored or reviewed drafts of the paper, approved the final draft.

Walter G. Joyce analyzed the data, prepared figures and/or tables, authored or reviewed drafts of the paper, approved the final draft.

The following information was supplied regarding data availability:

The raw data are provided in the Supplemental Files.

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
