# Peer review of "A taxonomic reassessment of Piramys auffenbergi, a neglected turtle from the late Miocene of Piram Island, Gujarat, India"

_PeerJ, doi:10.7717/peerj.5938_

## Round 0.1 · original submission · Minor Revisions

I have now received three reviews of your manuscript. I agree with all reviewers, that the study is very valuable but it needs a little more work. The reviewers provided several points that should be addressed in order to improve the ms. For the revised version of your work I would like you to explain the reasons why additional preparation, such as CT scans, or X-ray images were not undertaken. You need to expand upon the absence of nasals and its implications. Please add to your discussion section the paper of Rabi et al. (2012). It seems to me that the use of Africa and Asia a biogeographic units in the cladograms should be further discussed. Please add in the SM2 the source you used to change the scorings of the taxa. Additional recommendations of all reviewers are highly constructive, and attention to their suggestions will serve to improve the ms.

·

Basic reporting

This is a very clear and concisely written contribution on the taxonomic affinities of an isolated, and partially preserved turtle skull from the Miocene of Piram Island, India. The background and relevant literature are well covered. The authors re-studied the anatomy of the holotype skull concluding that it is a pleurodiran podocnemidid turtle, and not, as was originally proposed, a cryptodiran, i.e. a member of emydid turtles.

Experimental design

The authors use standard taxonomic methods to elucidate the status of the species. Their results are backed up by the data and by a number of support values to more confidently interpret their results. The primary research is also well within the scope of the journal and interesting to a broad audience of turtle workers.

As a large part of the anatomy of the specimen remains within sediment matrix, however, the reader is left wondering, whether additional preparation, CT scans, or even a simple X-ray image might not have revealed further anatomical features and thus added to the dataset used. The authors should thus at least include a sentence or two in the Materials and Methods section, why those additional steps were not undertaken.

Validity of the findings

Both the presented data in the main text as well as the supplementary file is deemed well presented and sound. The conclusions are well backed by the presented data.

The aspect on the absence of nasals and its implications needs to be expanded upon. For example in the "Abstract -Results section" (lines 22-23) it is stated that "absence of nasals" besides "the presence of a processus trochlearis pterygoidei" conclusively indicate pleurodiran affinities of the Piramys holotype skull. While I agree on the latter point, I am more cautious about the former - chelids, another prominent group of pleurodiran turtles, do have nasal bones, whereas most cryptodirans also lack nasals. This point thus needs to be refined here and in the discussion section.

In line 283, the statement "Given that all Asian taxa also form a clade..." needs to be refined, because they form a clade together with African taxa, as is later pointed out correctly in line 289.

Additional comments

I think this a very clear and concisely written contribution on the taxonomic affinities of an isolated, but only partially preserved turtle skull from the Miocene of Piram Island, India.
1) The authors should consider adding information to the Materials and Methods section as to why no further preparation, CT scanning or X-raying of the specimen was done to maximize the anatomical information that could be retrieved from the specimen.
2) The authors should consider revising the text at several part of the manuscript (Abstract, Results, Discussion) about the impact of the absence of nasals for their analysis.
3) I have added additional stylistic comments directly on the PDF.

I consider all my comments to be minor in nature.

·

Basic reporting

All necessary data is sufficently documented and the relevant literature is provided. If there is any ways to precise the locality that would be a welcome addition.

Experimental design

No need to be improved.

Validity of the findings

The conclusions are supported by the data and are balanced - a big bonus. I think using Africa and Asia as biogeographic units in the cladograms actually masks a monophyletic Tethyian clade. The phylogeny and occurrences suggest that this clade was widespread in the Tethys during the Paleogene. Is that bytheway correct that the outgroups to Stereogenynini do not suggest an earlier presence of the group than the Oligocene? Can you doublecheck? I think it makes less sense to classify these species as African or Asian since they are marine and were all around the Tethys. Perhaps use Paleogene marine provinces instead.

In a paper of mine describing a Cretaceous bothremydid (Rabi et al. 2012), I discuss reversal of nearshore marine to freshwater lifestyle in pelomedusoid turtles - this is perhaps relevant to the manuscript of the authors. I no longer think that what I proposed were actual reversals - for the same reasons as what the authors pointed out in their ms (euryhalininty of nearshore marine forms and transition of environments).

·

Basic reporting

The manuscript is clearly written, with clear and unambiguous, professional English. It is well-organized, with precise information, clear and concise. The references and background is well documented. I made some minor comments in the attached PDF. Besides, in the SM2 it would be good to add the source you used to change the scorings of the taxa.

Experimental design

The work is well-executed and fits in the aims and scope of this journal. The research is rigorous and using the appropriated methodology. The methodology is described in the way it can be reproduced.

Validity of the findings

The primary data (e.g., pictures, drawings, descriptions, phylogenetic analysis) is robust and clearly presented by the authors. Their findings are very interesting and bring to light some forgotten or poorly described taxa. The conclusions are well-supported by the data.

Additional comments

Nice job! Congratulations!

---

## Round 0.2 · Minor Revisions

I am glad with the modifications made to your manuscript. In relation to your statement about the presence of a pterygoid trochlea combined with the absence of nasals as an undisputable fact, references are needed to sustain your strong claim. Likewise, please add references sustaining that testudinoids universally lack nasal bones.

---

## Round 0.3 · accepted · Accept

Thank you very much for your consideration of all suggestions made to improve your manuscript. I think that it is ready to publish.

#